# The Effect of Perioperative Blood Transfusions on Microvascular Anastomoses

**DOI:** 10.3390/jcm10061333

**Published:** 2021-03-23

**Authors:** Lidia Sanchez-Porro Gil, Xavier Leon Vintro, Susana Lopez Fernandez, Carmen Vega Garcia, Gemma Pons Playa, Manuel Fernandez Garrido, Jaume Masia Ayala

**Affiliations:** 1Department of Plastic and Reconstructive Surgery, Hospital Santa Creu i Sant Pau, Universitat Autònoma de Barcelona, 08041 Barcelona, Spain; SLopezfe@santpau.cat (S.L.F.); CVega@santpau.cat (C.V.G.); GPonsP@santpau.cat (G.P.P.); MFernandezGa@santpau.cat (M.F.G.); jmasia@santpau.cat (J.M.A.); 2Department of Otolaryngology, Hospital de la Santa Creu i Sant Pau, Universitat Autònoma de Barcelona, 08041 Barcelona, Spain; xleon@santpau.cat

**Keywords:** microsurgery, transfusion, free flap, breast reconstruction

## Abstract

Introduction: Perioperative transfusions are associated with complications of free flaps. The purpose of the present study was to find out whether there is a significant relationship between the risk of developing complications in vascular anastomoses and the history of transfusions. Methods: We studied 372 patients retrospectively with microsurgical reconstruction between 2009 and 2017 with regards to the number of red blood cell concentrates transfused. Complications were analyzed relative to flap loss and complications in microvascular anastomoses. Results: 130 patients (34.9%) received blood transfusions. Some 55% of them were transfused between the day of the intervention and the first postoperative day. Ninety-six patients were reoperated on (25.7%). Of those, thirty-six patients (37.5%) corresponded to anastomosis failure. The percentage of patients transfused among those who required reoperation was 55.2%. The percentage of patients transfused among those who were reoperated on within the first 72 h due to an alteration in the anastomosis was 60.6%, while it was 25.6% (Chi square P = 0.0001) for the rest of the patients. Conclusions: Although there is a strong association between transfusion and vascular anastomosis failure, it is not possible to establish the causation between the two.

## 1. Introduction

The percentage of failures associated with the use of micro-anastomosed free flaps in the most recently published series stands at between 1% and 9% [1,2,3]. In addition to failure, which is understood as the partial or complete loss of the flap, the percentage of patients who suffered the appearance of complications during the perioperative period varied from one series to another.

Several studies have been carried out on large populations of patients treated with micro-anastomosed free flaps that analyze the risk factors associated with the appearance of postoperative complications and the failure of the reconstructive procedure. In those studies, the factors that are significantly associated with the loss of the flap are the need for an intraoperative transfusion of three or more units and an elevated surgical time. On the other hand, the relationship to other factors such as age, sex, BMI (body max index), smoking, a history of alcoholism, and the American Society of Anesthesiologist (ASA) score is not so clear [4,5].

During blood storage, the erythrocytes progressively undergo structural and biochemical changes. They become less deformable and more fragile, leading to the appearance of the hemolysis phenomena, the accumulation of pro-inflammatory molecules, iron, and free hemoglobin (Hb) [6]. The release of free Hb leads to a reduction in nitric oxide concentrations with consequent vasoconstriction [7], the induction of a state of hypercoagulability mediated by the adhesion of leukocytes, the increase in endothelial permeability, and the proliferation of smooth muscle after the induction of a vascular trauma [8]. It is estimated that after one hour of transfusion, up to 30% of the transfused red blood cells have hemolyzed or been removed by the action of macrophages [9]. Moreover, there is a depletion of 2,3-diphosphoglyceride acid (2,3 DPG) and adenosine triphosphate (ATP), altering the oxygen transport capacity [10]. All these modifications justify an increase in the appearance of the thrombotic phenomena associated with giving transfusions.

The main problem at the time of proposing a transfusion is to know to what extent hemoglobin (Hb) can decrease before provoking a dysfunction or severe tissue damage. Perioperative anemia contributes to an increase in morbidity and mortality. On the other hand, interventions like transfusions to correct this anemia also have adverse effects and produce complications that may increase the risk to patients. The benefit obtained with transfusions has a cost associated with the risk of bacterial or viral infections caused by the transfusion, immunological reactions including anaphylaxis and hemolysis, and disorders in the regulation of the immune system that could favor the appearance of infectious complications in the postoperative period [11]. From studies developed in hemodynamically stable patients without active blood loss admitted to critical care units, a transfusion threshold or trigger of 70 g/L has been established as reasonable to avoid the harmful effects of anemia [11,12].

It remains to be determined what the ideal Hb or hematocrit level is for patients treated with a micro-anastomosed free flap. Different studies have experimentally evaluated, in animal and human models, the effect of perioperative hematocrit on the microvascular success rate. They came up with contradictory results [5,13,14,15,16]. The effect of anemia on microvascular reconstruction is controversial. Some studies have suggested a benefit in normovolemic hemodilution based on the theory that the decrease in viscosity increases arterial flow and perfusion at the tissue level [15,16]. Other studies argue that basal anemia is a variable intrinsically related to the patient’s physiological state, but that it does not appear as a predictor of outcomes in patients treated with a micro-anastomosed free flap [17,18].

The main objective of the present study was to find out whether there is a significant relationship between the risk of developing complications in vascular anastomoses and the history of transfusion in reconstructions with micro-anastomosed free flaps.

## 2. Patients and Methods

The study included 372 patients with a history of microsurgical reconstruction in the Plastic Surgery service of the Hospital de la Santa Creu and Sant Pau between 2009 and 2017. Patients who had undergone a second microsurgical reconstruction were excluded from the study.

The number of red blood cell concentrates transfused was counted from seven days before the intervention until the fourth postoperative day. Likewise, the data referring to the preoperative and postoperative hemoglobin (Hb) values were collected.

The decision to transfuse the patient was made at the discretion of the anesthesiologist or surgeon in accordance with the protocol of the Hospital de la Santa Creu i Sant Pau. Transfusion is called for if Hb <70 g/L. However, if Hb is <80 g/L, transfusion is required in patients with difficulties adapting to anemia, those older than 65 years of age, and patients with a history of cardiovascular and/or pulmonary disease.

### 2.1. Parameters Evaluated

Clinical data on age, sex, BMI, vascular history, and preoperative and postoperative hemoglobin values were obtained from each patient. The transfusion requirements and the existence of reoperation and reason were studied: hematoma, anastomosis failure, or flap necrosis. The operative time was also studied. Complications have been classified as partial or total loss of the flap, wound dehiscence, fistula, or infection.

### 2.2. Statistical Analysis

A descriptive study was made of the characteristics of the patients, the results obtained, and the prognostic variables potentially related to the results. Depending on the conditions of application, the comparison between qualitative variables was done using the Chi-Square Test or Fisher’s exact test. The relationship between continuous and qualitative variables was evaluated using the Student’s t-Test for independent samples for the evaluation of dichotomous dependent variables and with the ANOVA test in the case of dependent variables with more than two categories. We classified patients based on transfusion requirements by recursive partitioning analysis (CHAID method), considering the failure of the anastomosis as a dependent variable.

## 3. Results

### 3.1. Overall Results

Of the 373 patients included in the study, 51% underwent reconstruction of the breast, 22% reconstruction of the head and neck, 20% reconstruction of the lower extremity, and the rest of the patients corresponded to a miscellany of several specialties. As for the flaps made, 97% corresponded to perforator flaps, 8% to osteocutaneous flaps, and the rest to non-perforating flaps. 

Ninety-six patients were reoperated on (25.7%). Of those, 45 (46.8%) corresponded to anastomosis failure, 19 (19.8%) to hematoma evacuation with no evidence of anastomosis failure, and 32 (33.3%) to debridement of partial flap. There were no significant differences in terms of reoperation in terms of the type of flap or the surgical indication. Most of the reinterventions, for revision of the anastomosis or hematoma evacuation, occurred during the first 72 h postoperatively (Figure 1).

There was a partial loss in 31 (8.3%) flaps even though they maintained their reconstructive purpose. In 24 (6.4%) instances, there was a total loss of the flap. Although a lower percentage of loss was observed in breast reconstructions, there were no significant differences as regards the surgical indication or the type of flap.

Of the 45 patients on whom revision of the anastomosis was performed, in 9 cases we performed a flap removal due to necrosis and in 36 patients we performed an anastomosis revision. Of the 36 patients on whom revision of the anastomosis was performed, there was a complete rescue of the flap in 14 (38.9%), partial flap loss in 8 (22.2%) without there being an alteration in the reconstructive purpose, and a total loss of the flap in 14 (38.9%). This means that repair of the anastomosis succeeded in 61.1% of the instances.

### 3.2. Transfusion Requirements

A total of 130 patients (34.9%) received blood transfusions. The highest density of transfusions occurred during the day of the intervention (intraoperative or immediate postoperative) and the first postoperative day. A total of 394 packed red blood cells were transfused. Some 55% of them were transfused between the day of the intervention and the first postoperative day. There were significant differences in the transfusion requirements relative to patient gender, it being more frequent in male patients (*p* = 0.001). There were also differences regarding the ASA score of the patients. There were no significant differences in terms of age or BMI.

The average Hb of the patients who did not receive a transfusion was 136 g/L, while the average of those who did receive a transfusion was 128 g/L (Figure 2). There was a significant relationship between preoperative Hb levels and the probability of receiving a blood transfusion (*p* = 0.0001). 

### 3.3. Relationship between Transfusion and Complications

The percentage of patients transfused among those who required reoperation was 55.2%, while it was 27% for patients who did not require reoperation (Table 1). The percentage of patients transfused among those who were reoperated on in the first 72 h due to an alteration in the anastomosis was 60.6%, while it was 25.6% (*p* = 0.0001) for the rest of the patients (Table 2). Of the total of 33 patients in whom the anastomosis was reviewed during the first 72 h after surgery, 15 (45.5%) had either undergone a prior transfusion or one on the same day of the revision. Transfusions were not required in 16 (48.5%) cases and transfusions were made after doing the revision of the anastomosis in 2 patients (6%) (Table 3).

Complete flap loss occurred 24 times (6.4%). Of the patients who suffered flap necrosis, 75% had received a transfusion, while the percentage was 32.1% for patients in whom there was no flap necrosis (Table 4).

The preoperative Hb of patients with flap loss was 125.5 g/L. The Hb of the patients who did not suffer loss of the flap was 134 g/L (*p* = 0.012). There was a significant relationship between the minimum perioperative Hb value and flap failure (90 g/L) and non-failure of the flap (98.4 g/L) (*p* = 0,012).

In a multivariate analysis, the transfused patients had a 6.28 times higher risk (CI 2.91–13.55) of requiring a surgical revision related to the vascular permeability of the micro-anastomosed flap. For each transfused concentrate, the risk of requiring a surgical reoperation due to problems with the permeability of the anastomosis increased 1.39 times (CI 1.20–1.61). No relationship was established between the preoperative Hb and surgical reoperation related to vascular permeability.

According to the results of a recursive partition analysis (CHAID method), three categories of patients were defined according to the transfusion needs. We considered failure at the anastomosis level as a dependent variable (Figure 3). The percentage of patients who required surgical reoperation as a consequence of a failure in the anastomosis was 4.9% for patients without transfusion (*n* = 243), 20.4% for patients with a transfusion of 1–3 concentrates, and 37.8% for patients with a transfusion of 4 or more concentrates.

### 3.4. Duration of Surgery

The average duration of surgery was 9.10 h (standard deviation 2.3 h, range 3.5–18 h). The reconstructive procedures at the level of the head and neck were those that had a longer average duration (10.46 h, SD: 2.24 h), while the breast reconstructions were those that had shorter average duration (8.52 h SD: 2 h)

A relationship between the duration of surgery and the use of transfusions (ANOVA *p* = 0.0001) could be seen. Patients who had not received transfusions had a mean surgery duration of 8.69 h while the average duration was 10.5 h in patients with four or more transfused concentrates.

## 4. Discussion

The main conclusion of the results is the existence of a significant relationship between the appearance of postoperative complications after performing a microsurgical reconstruction and the practice of transfusing. The percentage of patients transfused among those who required reoperation was 55.2%, while it was 27% for patients who did not require reoperation (Table 1). If only those reinterventions performed as a result of problems at the level of the anastomosis are considered, 60.6% of the patients had a history of perioperative transfusion (*p* = 0.0001). Equally, a significant relationship arose between the performance of transfusions and complete flap loss and the consequent reconstructive failure. The percentage of patients transfused among those in whom there was complete flap loss was 75%, while this percentage went down to 32.1% (*p* = 0.0001) for the rest of the patients. The totality of these findings indicates that there is a relationship between transfusion and complications. The majority of authors coincide in pointing out how transfusion during the perioperative period of microsurgery with free flaps increases the risk of medical or surgical complications and increases the duration of the hospital stay and the percentage of failure of the reconstructive procedure [4,5,17,18,19,20,21,22,23,24].

From the results obtained, we know that there is a relationship between transfusion and problems associated with the permeability of micro-anastomoses. However, we cannot determine to what degree this relationship is due to the transfusion itself, to the reasons why the transfusion was performed (like the appearance of anemia) or if the need for transfusion is the consequence of the alterations caused by the micro-anastomosis. To define the chronological relationship between the existence of problems at the level of the vascular anastomosis and the performance of transfusions, the sequence between giving a transfusion and surgical revision of the microvascular anastomosis was analyzed. Of the 33 patients in whom a revision of the anastomosis was performed, either a previous transfusion had been performed or it had been done on the same day on which the revision was carried out in 45.5% of the cases. It is a figure that is slightly higher than the 34.9% of patients that received some type of perioperative transfusion. In any case, we cannot guarantee that the realization of transfusions, especially those carried out on the same day as the surgical revision of the anastomosis, was before or after the complication for which the reoperation was carried out. Therefore, no type of causal relationship between transfusing and the appearance of complications can definitively be established.

Reconstructive failure, defined as the complete loss of the flap, affected 6.4% of the patients included in our series. This percentage is in the range of those previously reported in various studies, which calculated the percentage of necrosis or removal of the flap at between 2% and 9% [4,5,13,17,18,25].

As noted above, most authors have found a correlation between the occurrence of postoperative complications or reconstructive failure and the giving of transfusions during the perioperative period [4,5,17,18,19,20,21,22,24]. Other variables related to transfusion that have demonstrated an ability to influence the appearance of postoperative complications are the number of transfused concentrates [18] and the period of storage of blood prior to transfusion [23].

There are no studies in the literature that have been designed in such a way as to determine whether the appearance of complications is the reason for the transfusion being indicated, or if it is the transfusion itself that is related to the appearance of complications, especially in what refers to the permeability of the anastomoses. It is evident, for ethical reasons, that it is not possible to carry out randomized transfusion clinical trials regardless of their requirements in patients undergoing reconstructive surgery with free flaps. The only way to evaluate the causal relationship between transfusion and the appearance of problems with the permeability of vascular micro-anastomoses would be to carry out a prospective study with the ability to determine the chronological sequence between the permeability of anastomoses and giving transfusions.

## 5. Conclusions

Although there is a strong association between transfusion and vascular anastomosis failure, it is not possible to establish the causation between the two. Given the existing strong association, it would be convenient to avoid perioperative transfusion. The priority for microsurgeons should be to further minimize blood loss in the operating room and improve the management of perioperative anemia. To clarify the sequential relationship between transfusion and failure of anastomoses, it would be necessary to design a new, highly detailed prospective study.

## Figures and Tables

**Figure 1 jcm-10-01333-f001:**
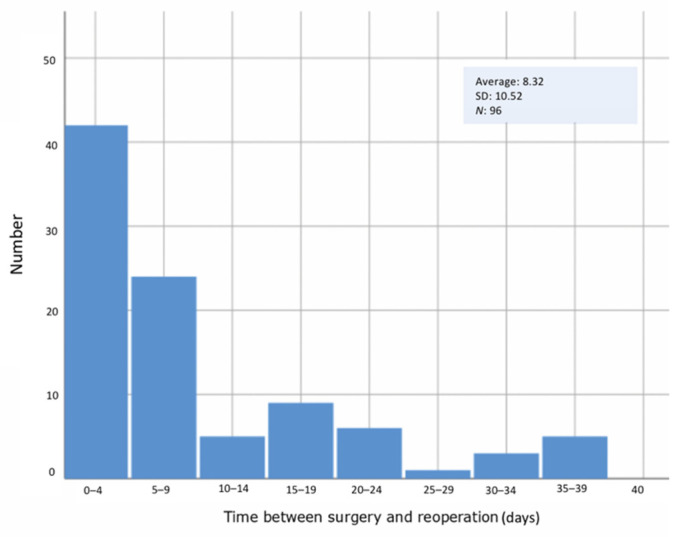
Interval between the day of the surgery and reoperation.

**Figure 2 jcm-10-01333-f002:**
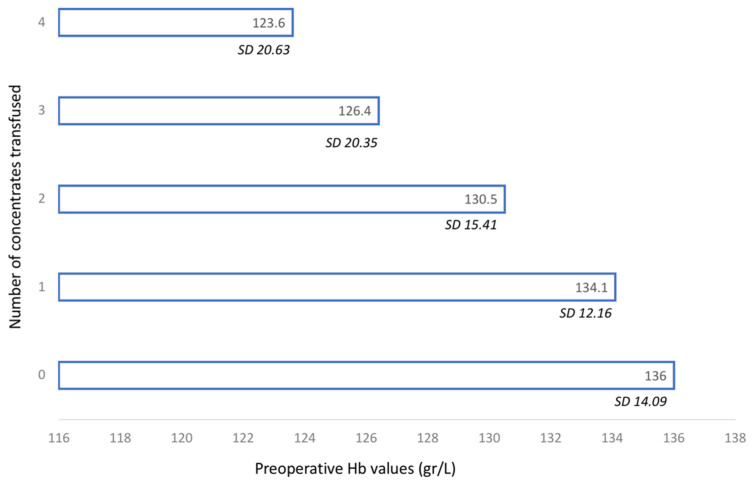
Preoperative hemoglobin values (g/L) in relation to the number of transfused blood concentrates (0–4).

**Figure 3 jcm-10-01333-f003:**
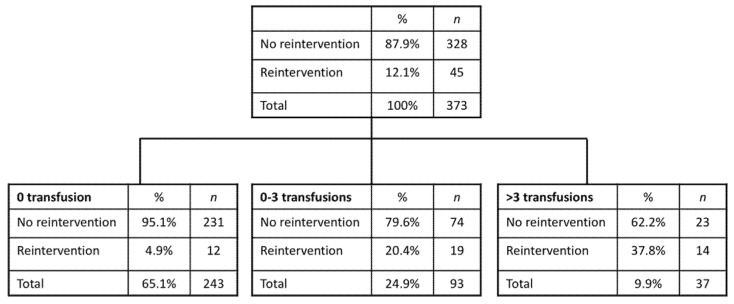
Recursing partitioning analysis (RPA) was used to create a regression tree according to reoperation rates and transfusion.

**Table 1 jcm-10-01333-t001:** Relationship between blood transfusion and reoperated patients.

		Blood Transfusion		Total
		No	Yes	
Reoperation	No	200 (72.2%)	77 (27.8%)	277 (100.0%)
	Yes	43 (44.8%)	53 (55.2%)	96 (100.0%)
Total		243 (65.1%)	130 (34.9%)	37 3(100.0%)

**Table 2 jcm-10-01333-t002:** Relationship between reoperations due to anastomosis failure and blood transfusion during the first 48 h.

		Blood Transfusion in the First 48 h		Total
		No	Yes	
Anastomosis failure	No	253 (74.4%)	87 (25.6%)	340 (100.0%)
	Yes	13 (39.4%)	20 (60.6%)	33 (100.0%)
Total		266 (71.3%)	107 (28.7%)	373 (100.0%)

**Table 3 jcm-10-01333-t003:** Relationship between the day of the first transfusion and the day of the anastomosis revision.

		First Transfusion				Total
		No	Day Zero	Day One	Day Two	
Anastomosis revision	Day zero	2 (6.1%)	0 (0%)	0 (0%)	0 (0%)	2 (6.1%)
	Day one	9 (27.3%)	2 (6.1%)	8 (24.2%)	2 (6.1%)	21 (63.6%)
	Day two	2 (6.1%)	0 (0%)	3 (9.1%)	1 (3.0%)	6 (18.2%)
	Day three	3 (9.1%)	0 (0%)	1 (3.0%)	0 (0%)	4 (12.1%)
Total		16 (48.5%)	2 (6.1%)	12 (36.4%)	3 (9.1%)	33 (100.0%)

**Table 4 jcm-10-01333-t004:** Relationship between transfusion and flap loss.

		Transfusion		Total
		No	Yes	
Flap loss	No	237 (67.9%)	112 (32.1%)	349 (100%)
	Yes	6 (25.0%)	18 (75.0%)	24 (100%)
Total		243 (65.1%)	130 (34.9%)	373 (100%)

## Data Availability

The statistical study was carried out with the SPSS 17.0 program.

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
