# Peer review of "The Effect of Perioperative Blood Transfusions on Microvascular Anastomoses"

_jcm, 2021, doi:10.3390/jcm10061333_

Round 1

Reviewer 1 Report

The article is a retrospective study of 372 patients who undergone microsurgical reconstruction, evaluating the effect of blood transfusion on complication. The authors find a relationship between transfusion and vascular anastomosis failure. However, it's impossible to establish whether the increase of complications is related to transfusions or the patient's clinical picture that indicated the transfusion. 

The topic of the article has already been studied by other authors but it is still discussed and therefore still interesting. The new analysis performed by the authors is the chronological relationship between giving a transfusion and surgical revision of the microvascular anastomosis.

The article conforms to the journal-specific instructions. The article is well written, the statistical analysis is solid and the graphics explanatory. The results are clearly presented and the conclusions are supported by the results.

Author Response

Dear colleague,

Thank you very much for your kind review

Reviewer 2 Report

Submit peer-review report for manuscript jcm-1127821

The authors present their experience in transfusion in 372 consecutive free tissue transfer.  The authors have a very nice experience in microsurgery and appear to sensibly manage blood loss in a very diverse group of difficult cases. 

I have several suggestions for improvement that the authors may consider prior to publication:

  • As a retrospective cohort study, attaching a STROBE checklist would assure reviewers that all expected elements in the manuscript are present.
  • The takeback and failure rates of their series are higher than other published series. The authors may wish to comment on this (more difficult cases or better data requirements).   It would be helpful to the readers to learn a bit more about the flaps that failed.  Perhaps they can include a table in supplementary data listing the flaps that failed and were taken back.
  • The authors basically are showing associations between data elements.

For me, the word “association” is clearer than the word “relationship”.  Similarly, I would have a preference for the word “causation” rather than “sequential link”

  • The authors have gathered a large amount of data. They could consider multiple logistic regression analysis as a way of better modeling their data. 
  • I appreciate the authors publishing in English, and would encourage them to have this reviewed by a native speaker. In addition, there are several instances of persistent Spanish across tables and figures.    
  • There are excess significant digits in the tables. For less than 1000 subjects, only 3 significant digits should be employed.  For example: “8.3229” would better be reported as “8.32”
  • Figure 1 needs units on both axes. The Y-axis would be better labeled as “Number”, on the X-axis, ideally a range between each bar would be better, for example “0 – 5” rather than “5”
  • For Figure 2, standard deviations should be added to the bars.
  • Tables 1 and 3 are bolded in some spots. 
  • Some of the figures appear to be direct printouts from a statistical program – professionally done graphics would be very helpful and easier to read.
  • Why wasn't surgical blood loss included or recorded during the study?  If the reason is these numbers are often inaccurate because based on estimates, it might be worth stating.  
  • I found Table 1 confusing as it did not specify the timing of the transfusions in relationship with the reoperation. 

Thank you for reporting your experience. 

Author Response

Dear colleague,

I appreciate your corrections, they helped to improve the quality of the manuscript.

I will answer every point of your review:

  1. As a retrospective cohort study, attaching a STROBE checklist would assure reviewers that all expected elements in the manuscript are present. DONE. I attatch a file.
  2. The takeback and failure rates of their series are higher than other published series. The authors may wish to comment on this (more difficult cases or better data requirements).   It would be helpful to the readers to learn a bit more about the flaps that failed.  Perhaps they can include a table in supplementary data listing the flaps that failed and were taken back. RESPONSE: Although it is true that we work in a leading hospital in surgery for sarcomas and head and neck tumors, with more limited patients and in worse basic conditions, the percentatge of reconstructive failure defined as loss of the flap is 6,4%, this percentage is in the range of other reports (references 4,5,13, 18,25). As the objective of the study is the association between transfusion and loss of flap, we have not considered it useful to include the list of reoperated flaps
  3. For me, the word “association” is clearer than the word “relationship”.  Similarly, I would have a preference for the word “causation” rather than “sequential link”. RESPONSE: changes made
  4. The authors have gathered a large amount of data. They could consider multiple logistic regression analysis as a way of better modeling their data. RESPONSE: Although the colleague in charge of analyzing the statistical data used the tests that he considered appropriate, I appreciate his contribution to be able to improve our studies in future scientific works.
  5. I appreciate the authors publishing in English, and would encourage them to have this reviewed by a native speaker. In addition, there are several instances of persistent Spanish across tables and figures.   RESPONSE: changes is tables and figures done. The text has been reviewed again by a native speaker. Thank you, we try our best to communicate in English.
  6. There are excess significant digits in the tables. For less than 1000 subjects, only 3 significant digits should be employed.  For example: “8.3229” would better be reported as “8.32”. RESPONSE: changes made.
  7. Figure 1 needs units on both axes. The Y-axis would be better labeled as “Number”, on the X-axis, ideally a range between each bar would be better, for example “0 – 5” rather than “5”. RESPONSE: changes made.
  8. For Figure 2, standard deviations should be added to the bars. RESPONSE: changes made
  9. Tables 1 and 3 are bolded in some spots. RESPONSE: changes made
  10. Some of the figures appear to be direct printouts from a statistical program – professionally done graphics would be very helpful and easier to read.RESPONSE: new figure made
  11. Why wasn't surgical blood loss included or recorded during the study?  If the reason is these numbers are often inaccurate because based on estimates, it might be worth stating.  RESPONSE: We have not taken into account blood loss because, as it is a retrospective study, it is difficult to collect the data. Likewise, we have considered it more appropriate to take the hemoglobin value as a target value for postoperative blood loss. It would be interesting in the case of designing a new prospective study
  12. I found Table 1 confusing as it did not specify the timing of the transfusions in relationship with the reoperation. RESPONSE: In Table 1 we can see the association between transfusion and patients who have had to be reoperated. Table 3 shows the association between the days elapsed since the intervention and the day of the transfusion. I understand that the table may be difficult to understand, the text is intended to complement the explanation for better understanding.

Thank you very much for reading the manuscript carefully in order to improve its scientific quality.

STROBE CHECKLIST

Item No

Recommendation

 Title and abstract

1

(a) Indicate the study’s design with a commonly used term in the title or the abstract

Page 1

(b) Provide in the abstract an informative and balanced summary of what was done and what was found

Page 1

Introduction

Background/rationale

2

Explain the scientific background and rationale for the investigation being reported

Page 1-2

Objectives

3

State specific objectives, including any prespecified hypotheses

Page 2

Methods

Study design

4

Present key elements of study design early in the paper

Page 2

Setting

5

Describe the setting, locations, and relevant dates, including periods of recruitment, exposure, follow-up, and data collection

Page 2

Participants

6

(a) Give the eligibility criteria, and the sources and methods of selection of participants. Describe methods of follow-up

Page 2

(b) For matched studies, give matching criteria and number of exposed and unexposed

N/A

Variables

7

Clearly define all outcomes, exposures, predictors, potential confounders, and effect modifiers. Give diagnostic criteria, if applicable

N/A

Data sources/measurement

8*

 For each variable of interest, give sources of data and details of methods of assessment (measurement). Describe comparability of assessment methods if there is more than one group

N/A

Bias

9

Describe any efforts to address potential sources of bias

Study size

10

Explain how the study size was arrived at

All the patients operated included

Quantitative variables

11

Explain how quantitative variables were handled in the analyses. If applicable, describe which groupings were chosen and why

Statistical methods

12

(a) Describe all statistical methods, including those used to control for confounding

Page 3

(b) Describe any methods used to examine subgroups and interactions

Page 3

(c) Explain how missing data were addressed

(d) If applicable, explain how loss to follow-up was addressed

N/A

(e) Describe any sensitivity analyses

Results

Participants

13*

(a) Report numbers of individuals at each stage of study—eg numbers potentially eligible, examined for eligibility, confirmed eligible, included in the study, completing follow-up, and analysed

(b) Give reasons for non-participation at each stage

N/A

(c) Consider use of a flow diagram

Descriptive data

14*

(a) Give characteristics of study participants (eg demographic, clinical, social) and information on exposures and potential confounders

Page 3 (types of reconstruction)

(b) Indicate number of participants with missing data for each variable of interest

(c) Summarise follow-up time (eg, average and total amount)

Postopeartive time

Outcome data

15*

Report numbers of outcome events or summary measures over time

Main results

16

(a) Give unadjusted estimates and, if applicable, confounder-adjusted estimates and their precision (eg, 95% confidence interval). Make clear which confounders were adjusted for and why they were included

Page 3

(b) Report category boundaries when continuous variables were categorized

Page 4 (Hb and blood cells transfusions)

(c) If relevant, consider translating estimates of relative risk into absolute risk for a meaningful time period

N/A

Other analyses

17

Report other analyses done—eg analyses of subgroups and interactions, and sensitivity analyses

Page 5 (recursive partition analysis -CHAID method)

Discussion

Key results

18

Summarise key results with reference to study objectives

Page 4-5

Limitations

19

Discuss limitations of the study, taking into account sources of potential bias or imprecision. Discuss both direction and magnitude of any potential bias

Page 7

Interpretation

20

Give a cautious overall interpretation of results considering objectives, limitations, multiplicity of analyses, results from similar studies, and other relevant evidence

Page 6-7

Generalisability

21

Discuss the generalisability (external validity) of the study results

Page 7

Other information

Funding

22

Give the source of funding and the role of the funders for the present study and, if applicable, for the original study on which the present article is based

Page 7- no external funding
